# Telehealth Perceived Benefits and Self-Efficacy Do Not Mediate the Effects of Demographic, Health, and Social Determinants on Telehealth Use of Low-Income African American and Latino Residents of Public Housing in Los Angeles

**DOI:** 10.3390/healthcare13030286

**Published:** 2025-01-31

**Authors:** Sharon Cobb, Attallah Dillard, Ehsan Yaghmaei, Mohsen Bazargan, Shervin Assari

**Affiliations:** 1Mervyn M. Dymally College of Nursing (MMDCON), Charles R. Drew University of Medicine and Science (CDU), Los Angeles, CA 90059, USA; sharoncobb1@cdrewu.edu (S.C.); atalahdillard@cdrewu.edu (A.D.); 2Department of Family Medicine, Charles R. Drew University of Medicine and Science (CDU), Los Angeles, CA 90059, USA; ehsanyaghmaei@cdrewu.edu; 3Department of Internal Medicine, Charles R. Drew University of Medicine and Science (CDU), Los Angeles, CA 90059, USA; mohsenbazargan@cdrewu.edu; 4Department of Public Health, Charles R. Drew University of Medicine and Science (CDU), Los Angeles, CA 90059, USA

**Keywords:** telehealth, social determinants of health, racial and ethnic minorities, public housing, intersectionality, health disparities, mediation analysis

## Abstract

Background: Marginalized low-income racial and ethnic minority residents of public housing represent an intersectional population with multiple health needs and low healthcare utilization. Telehealth has been proposed as a solution to address healthcare access disparities, yet the role of telehealth attitudes, including perceived benefits and self-efficacy, in influencing telehealth use of this population remains unknown. Objective: This study investigates whether two domains of telehealth attitudes, namely self-efficacy and perceived benefits (expectancy), mediate the relationship between demographic, health, and social determinants of telehealth use among low-income racial and ethnic minority residents of public housing in Los Angeles. Methods: This cross-sectional study analyzed data collected from low-income racial and ethnic minority residents of public housing in Los Angeles. Measures included demographic factors (age and gender), social determinants of health (e.g., education, language, and primacy care provider), health (chronic illnesses and physical mobility), telehealth attitudes (perceived benefits and self-efficacy), and telehealth use frequency. Mediation analysis was conducted to test whether telehealth attitudes explained the association between demographic, social, and health determinants of telehealth use. Results: The findings revealed that neither of the two domains of telehealth attitudes, including perceived benefits and self-efficacy, were significantly associated with telehealth use. As such, these attitude domains did not operate as mediators of the relationship between demographic, health, and social determinants of telehealth use. Conclusion: The study results suggest that structural barriers, rather than individual attitudes, primarily drive telehealth use disparities among this intersectional population. Interventions aimed at increasing telehealth adoption should prioritize addressing systemic inequities rather than focusing solely on changing individual attitudes. These findings underscore the importance of structural solutions to promote equitable telehealth access in marginalized communities.

## 1. Background

Racial and ethnic minority residents of public housing represent the intersection of three distinct social and contextual dimensions [1]: low-income status, racial and ethnic minority identity, and residency in public housing. Each of these factors independently shapes individuals’ experiences, resources, and access to healthcare, including telehealth [2]. However, their intersection creates unique conditions that cannot be fully understood by examining any single factor in isolation [3,4,5,6]. This aligns with the intersectionality theory and framework [7,8], which emphasizes how overlapping social identities and systemic inequalities create compounded effects. Despite the growing importance of telehealth, limited research has focused on the attitudes and usage patterns of this intersectional population, leaving a significant gap in understanding how these combined identities influence telehealth utilization and the potential barriers they face. This gap underscores the need for targeted studies and interventions to address the specific challenges and opportunities within this community.

Racial and ethnic minority residents of public housing often exhibit lower rates of healthcare utilization despite experiencing multiple chronic health conditions and complexing health needs [9,10,11]. Structural barriers such as financial constraints, limited access to transportation, and systemic discrimination in healthcare settings contribute to their reduced healthcare use [12,13]. Commonly centralized in urban residential cities, public housing areas have faced social and racialized patterns of residential segregation for decades, leading to systemic socioeconomic disparities [14,15,16]. Therefore, availability and access to health care facilities and providers may be increasingly lower in communities surrounding public housing areas [17,18]. Residents in public housing may be unlikely to find a primary or specialty provider within the local community, such as mental health or cardiology [19,20,21]. The mismatch between their elevated unmet health needs and lower healthcare utilization usage highlights the urgency of addressing the systemic and structural inequities that impede their ability to access and benefit from healthcare, including emerging modalities like telehealth.

Racial and ethnic minorities, specifically African Americans and Latinos, in low-income areas are likely to live in under-resourced and underserved areas and face extreme disparities in their healthcare seeking behaviors and encounters [22]. African Americans and Latinos exhibit low levels of healthcare utilization, including mental health [23]. Additionally, African American and Latinos may delay their care, which resulted in a study of 33 Latinas discussing treatment seeking behaviors of urinary incontinence [24]. Research shows minorities are more likely to rate printed materials as their preferred source of health communication compared to smartphones and tablets [25]. Yet, minorities who utilize the internet for health information are likely to have a higher education level and worse health status [26,27], despite its positive benefits for this group [28].

Telehealth presents a promising solution to address the healthcare access challenges faced by racial and ethnic minority residents of public housing [29]. By reducing logistical barriers such as transportation and providing more flexible options for care, telehealth has the potential to bridge the gap between their multiple health needs and low healthcare utilization. However, the effectiveness of telehealth adoption within this population may be influenced by their attitudes toward this modality [30]. Additionally, marginalized minorities are more likely to utilize telehealth via smartphone as compared to a computer [31]. The Health Belief Model [32,33] suggests that perceived benefits of telehealth, such as convenience and accessibility, could play a critical role in shaping behavior. Similarly, the Theory of Reasoned Action [34,35] highlights the importance of perceived self-efficacy in adopting telehealth services, where individuals’ confidence in their ability to use and benefit from the technology may influence their engagement. While these theoretical frameworks provide valuable insights, it remains unclear whether these associations are observed among this intersectional population.

Perceived benefits of telehealth can encompass improved patient health, positive feelings of empowerment and support, efficient communication, and greater patient education [36,37]. Telehealth self-efficacy is focused on the individual’s belief that they have the ability to perform the necessary actions to overcome barriers toward telehealth use and adoption and increase confidence [38,39,40,41,42]. More research is needed to understand whether and how telehealth attitudes, including perceived benefits [43,44,45,46] and self-efficacy [47,48,49,50], influence actual telehealth use in this group of public housing residents.

Informed by the Health Belief Model and the Theory of Reasoned Action, the overall aim of this study is to investigate whether telehealth perceived benefits and self-efficacy mediate the relationship between social determinants of health and telehealth utilization among African American and Latino public housing residents in Los Angeles. As a secondary aim, this study seeks to examine the gap in empirical evidence on how social, economic, and contextual factors influence telehealth utilization. Ultimately, the findings of this research will inform interventions and policies to enhance equitable access to telehealth services for marginalized communities.

## 2. Methods

### 2.1. Study Design

This cross-sectional study utilized a structured, face-to-face survey for data collection among African American and Latino residents of public housing areas in Los Angeles.

### 2.2. Participants

A total of 252 African American and Latino adults residing in four public housing areas located in South Los Angeles, California, were included in this study. Data collection spanned between August 2023 and January 2024. Out of 14 public housing sites managed by the Housing Authority of the City of Los Angeles (HACLA), 4 sites were selected, comprising a total of 2306 housing units. HACLA, one of the oldest public housing authorities in the United States, primarily oversees multiple government-controlled and affordable housing units in the Los Angeles region. As the issuer of housing for low-income individuals, HACLA currently accommodates 118,554 individuals across 6058 units. The average family income of those served by HACLA is substantially less than both the county and national averages (USD 24,881, compared to USD 68,044 and USD 62,843, respectively). Notably, more than 75% of HACLA residents self-identify as Latino, while 22% identify as African American/Black. The four selected housing sites are located within Service Planning Area (SPA) 6 of Los Angeles County, a region that is home to over one million residents and is marked by pronounced health disparities compared to other areas within the county.

### 2.3. Sampling

Participants were recruited through nonrandom sampling. Eligibility criteria including residency in SPA 6, self-identification as Latino or African American, and the ability to complete the interview in either English or Spanish. Adults aged 18 years or older were invited to participate, with eligibility for public housing determined based on income, age, disability status, and U.S. citizenship or immigration status.

### 2.4. Ethics

All participants provided written informed consent, and the study protocol received approval from the Institutional Review Board (IRB) at Charles R. Drew University of Medicine and Science. The survey collected detailed data on demographic characteristics, socioeconomic status, access to and utilization of healthcare services, health behaviors, and overall health status.

### 2.5. Measurement

A range of demographic, socioeconomic, and health-related data were collected from the participants. Basic demographic information included age and gender. *Age* was treated as a continuous variable ranging from 18 to 77 years. *Gender*, based on self-identification, was categorized as a dichotomous variable: (1) Female and (2) Male.

#### 2.5.1. Socioeconomic Status

We measured their socioeconomic status by the following variables: *Educational Level,* assessed by self-report based on the highest level of education completed, which was categorized into five levels: (1) 8th grade or less, (2) some high school, no diploma, (3) high school graduate or GED equivalent, (4) associate degree or some college, and (5) baccalaureate degree or higher.

*Primary verbal language*, which was measured as a dichotomous variable: (1) English, and (2) Spanish.

*Chronic Conditions*: participants reported if they have been diagnosed with any of the following conditions: cancer, chronic obstructive pulmonary disease (COPD), diabetes mellitus, cardiovascular diseases, hypertension, sleep disorders, and stroke. The number of chronic conditions was measured on a scale from 0 to 8.

*Physical mobility* was assessed using four items from the health-related quality of life survey (SF-12), which evaluated the extent to which participants’ health restricted their physical mobility. Participants were asked if their health limited them in moderate activities (e.g., moving a table, pushing a vacuum cleaner, bowling, or playing golf) and climbing several stairs, and, if so, to what extent. Participants also reported how often their physical health caused them to accomplish less than they desired and if there were limitations with their daily activities over the past four weeks. A total “Physical Mobility Limitation” score was calculated by summing these four items, with scores ranging from 4 to 12; higher scores indicated fewer limitations in physical mobility. The Cronbach’s alpha reliability coefficient for this 4-item scale was 0.820.

Participants were asked if they have a regular doctor or healthcare provider for primary care. This variable was dichotomous, with responses categorized as follows: (0) no and (1) yes.

#### 2.5.2. Telehealth

*Telehealth utilization:* Telehealth usage incorporated participants using one of the following devices to communicate virtually with their provider for a medical visit: computer, smartphone or telephone, or tablet. We assessed two aspects of attitudes toward telehealth utilization by utilizing questions and statements from previously established surveys on telehealth perceptions. Specifically, we incorporated items from recognized resources to measure responses to True/False statements concerning self-efficacy and perceived benefits of telehealth. The aggregated responses produced quantitative scores with higher scores indicating a more positive attitude toward telehealth [8,9]. The Cronbach’s alpha reliability coefficients for both of these domains were above 0.70. The outcome was the frequency of telehealth use in the past 12 months. The specific question read as “How many times have you used telehealth in the past 12 months?”.

*Telehealth perceived benefits:* Telehealth perceived benefits refer to an individual’s recognition of the advantages of using telehealth services, such as increased accessibility, convenience, reduced travel time, and cost savings. Patients and providers who perceive telehealth as beneficial are more likely to adopt and utilize it for routine and specialized care. Additionally, perceived benefits extend beyond logistical convenience to include improved health monitoring, timely interventions, and better management of chronic conditions, particularly for individuals in remote or underserved areas.

*Telehealth self-efficacy*: Telehealth self-efficacy describes an individual’s confidence in their ability to effectively use telehealth technology for healthcare needs. This construct encompasses skills such as navigating telehealth platforms, troubleshooting technical issues, and communicating effectively with healthcare providers in a virtual setting. Higher self-efficacy is associated with greater engagement in telehealth services, as individuals who feel competent in using these tools are more likely to participate in virtual consultations, follow medical advice, and integrate telehealth into their routine healthcare management.

### 2.6. Statistical Analysis

Our analysis comprised three sections. First, we conducted a descriptive analysis of all participants. Second, we used Pearson correlation coefficients to examine the bivariate association between all study variables including social determinants of health, telehealth perceived benefits and self-efficacy, and telehealth utilization within the past year and all other independent variables. Third, we employed a structural equation model (SEM) to evaluate the association between social determinants of health, telehealth attitudes, and the frequency of telehealth visit attendance. The analysis included demographic factors (age and gender), socioeconomic status (education), language, physical mobility, number of chronic conditions, and having a regular doctor or healthcare provider as predictors, positive attitude towards telehealth care utilization as mediators, and telehealth use as the outcome.

## 3. Results

Table 1 presents the characteristics of the study sample. This study comprised 252 African American and Latino individuals aged between 18 and 77 years (mean = 45 ± 15). Approximately 75% of the participants were female. Regarding educational attainment, 39% of participants reported never completing high school, 22% completed high school, 26% graduated from high school or attained a GED, and the remaining 12% held an associate degree or had some college education. Furthermore, 82% of the participants had a regular doctor/healthcare provider for primary care, and 74% indicated Spanish as their preferred language. At least 19% reported one chronic condition, whereas 15% had two chronic conditions, and 18% had three or more chronic conditions. The average physical mobility limitation score was 9 with a standard deviation of 2, indicating a lower level of physical mobility limitation within the sample. Regarding the use of telehealth services in the past 12 months, the data indicate that the majority of participants have not used telehealth services. With regard to telehealth utilization, 53% reported no telehealth usage within the past year, 20% participated once, 15% completed a telehealth visit twice, and 12% reported at least three telehealth visits.

Table 2 shows bivariate correlations between the use of telehealth services in the past 12 months and other study variables. Findings revealed that age and education were not correlated with telehealth utilization. However, gender (r = −0.138; *p* < 0.05), language (r = −0.151; *p* < 0.05), physical mobility limitation (r = −0.220; *p* < 0.001), and number of chronic diseases (r = 0.196; *p* < 0.001) were associated with telehealth utilization. In addition, having a primary care provider (r = 0.196; *p* < 0.001) was significantly associated with telehealth utilization.

The structural equation model results (Table 3, Figure 1) showed that telehealth expectancy was not significantly associated with telehealth utilization (B = 0.102, SE = 0.090, 95% CI = −0.075 to 0.280, *p* = 0.258). Similarly, telehealth self-efficacy did not demonstrate a significant relationship with telehealth utilization (B = 0.142, SE = 0.091, 95% CI = −0.036 to 0.320, *p* = 0.118). Among the covariates, age (B = −0.028, SE = 0.071, 95% CI = −0.168 to 0.111, *p* = 0.691) and education (B = −0.028, SE = 0.076, 95% CI = −0.177 to 0.121, *p* = 0.712) were not significant predictors of telehealth utilization. Gender, however, was significantly associated with telehealth use, with men reporting lower telehealth utilization compared to women (B = −0.128, SE = 0.059, 95% CI = −0.243 to −0.013, *p* = 0.029). Language preference also showed significance, with Spanish speakers reporting less frequent telehealth use (B = −0.135, SE = 0.069, 95% CI = −0.270 to 0.000, *p* = 0.050). Having a regular provider was positively associated with telehealth utilization (B = 0.121, SE = 0.059, 95% CI = 0.005 to 0.237, *p* = 0.041), and chronic disease status approached significance (B = 0.125, SE = 0.066, 95% CI = −0.004 to 0.254, *p* = 0.058). Physical fitness was inversely related to telehealth utilization (B = −0.149, SE = 0.065, 95% CI = −0.276 to −0.022, *p* = 0.021).

The results indicate that education significantly predicted telehealth self-efficacy, with higher levels of education associated with greater self-efficacy (B = 0.172, SE = 0.079, 95% CI = 0.018 to 0.326, *p* = 0.029). None of the other predictors, including age (B = 0.104, SE = 0.075, 95% CI = −0.043 to 0.251, *p* = 0.164), gender (B = −0.038, SE = 0.062, 95% CI = −0.160 to 0.084, *p* = 0.545), language preference (Spanish speakers: B = −0.112, SE = 0.073, 95% CI = −0.254 to 0.031, *p* = 0.126), having a regular doctor (B = 0.046, SE = 0.063, 95% CI = −0.077 to 0.170, *p* = 0.463), chronic disease (B = −0.047, SE = 0.070, 95% CI = −0.184 to 0.090, *p* = 0.504), or physical fitness (B = −0.005, SE = 0.069, 95% CI = −0.140 to 0.131, *p* = 0.947), significantly predicted self-efficacy.

For telehealth expectancy, none of the predictors showed significant associations. Age (B = 0.060, SE = 0.076, 95% CI = −0.089 to 0.208, *p* = 0.432), gender (B = −0.080, SE = 0.063, 95% CI = −0.203 to 0.043, *p* = 0.201), education (B = 0.048, SE = 0.080, 95% CI = −0.109 to 0.205, *p* = 0.548), language preference (Spanish speakers: B = −0.132, SE = 0.074, 95% CI = −0.276 to 0.013, *p* = 0.074), having a regular doctor (B = 0.056, SE = 0.064, 95% CI = −0.069 to 0.180, *p* = 0.381), chronic disease (B = 0.016, SE = 0.071, 95% CI = −0.123 to 0.154, *p* = 0.826), and physical fitness (B = −0.012, SE = 0.070, 95% CI = −0.149 to 0.125, *p* = 0.867) were not significant predictors of expectancy.

## 4. Discussion

This study examined the associations between socioeconomic status, health factors, telehealth perceived beliefs and self-efficacy, and telehealth utilization among African American residents within Los Angeles public housing areas.

### 4.1. Highlights from the Study

The findings revealed that telehealth attitudes, specifically perceived beliefs and self-efficacy, did not mediate the relationship with telehealth utilization. Instead, sociodemographic and health factors, such as gender, language preference, access to a regular doctor, and physical fitness, emerged as significant predictors of telehealth use. Additionally, education was the only significant predictor of telehealth self-efficacy, while no factors were significantly associated with telehealth perceived beliefs.

The absence of an effect of education on telehealth expectancy or utilization (frequency) can be explained by the theory of Marginalization-Related Diminished Returns (MDRs) [51]. According to this theory, factors such as limited resources in the social context, low-quality education, segregation, and other aspects of social marginalization weaken the real-life impact of socioeconomic status (SES) indicators, such as education, for underserved populations [52].

The results highlight the role of demographic, socioeconomic, and health factors in telehealth utilization. Women were more likely to use telehealth services, consistent with prior research suggesting gendered patterns in healthcare-seeking behavior [53,54,55,56,57]. Across studies and settings, women show more frequent healthcare use [53,54,55,56,57]. Language preference also played a significant role, with Spanish speakers reporting lower telehealth use. Not being able to speak English fluently in an English-speaking country is shown to be a barrier against health care use [58,59,60,61]. This finding underscores the need for culturally and linguistically tailored telehealth interventions to reduce barriers for non-English-speaking populations [62,63,64]. Furthermore, access to a regular doctor was positively associated with telehealth use, suggesting that existing relationships with healthcare providers may facilitate telehealth adoption [65,66,67,68]. Conversely, better physical fitness was linked to lower telehealth utilization, potentially reflecting a lower perceived need for medical care among healthier individuals.

One of the major findings of this study was that both telehealth attitudes have very limited predictive value for telehealth care utilization. While telehealth attitudes, including expectancy and self-efficacy, are often theorized to influence behavior, our findings suggest these attitudes did not significantly predict telehealth use in this intersectional population. This discrepancy may reflect the dominance of structural and systemic barriers in shaping telehealth behaviors, overshadowing the potential role of individual attitudes. The lack of significant predictors for telehealth expectancy further suggests that structural inequities may limit the extent to which individuals perceive benefits from telehealth services. On the other hand, the significant association between education and telehealth self-efficacy aligns with existing theories, such as the Health Belief Model and the Theory of Reasoned Action, which emphasize the role of education in shaping confidence and the perceived ability to engage with health technologies.

### 4.2. Implications

The findings have significant implications for addressing healthcare disparities through telehealth. Interventions should prioritize dismantling structural barriers, such as social segregation and improving language access and increasing provider availability. For instance, expanding access to bilingual telehealth platforms and fostering trust-based relationships between under-resourced populations and healthcare providers could play a pivotal role in enhancing telehealth adoption. Additionally, targeted efforts to engage men and individuals with limited access to regular care are essential to ensuring equitable telehealth use. Interventions that overly emphasize improving individual-level attributes, such as attitudes toward telehealth, are unlikely to bring about meaningful behavioral change. Effective solutions must go beyond individual attitudes and address systemic and structural factors to achieve lasting impact.

### 4.3. Study Limitations

This study has several limitations that should be acknowledged. First, the cross-sectional design limits the ability to draw causal inferences about the relationships between social determinants, telehealth attitudes, and telehealth use. While the findings provide valuable insights, they only capture a snapshot in time, leaving the temporal dynamics and causal pathways unexplored. Second, this study was conducted in Los Angeles using a non-random, convenience sample, which limits the generalizability of the results to other regions or populations. The unique socioeconomic, cultural, and healthcare landscape of Los Angeles may have influenced the findings, and different results could emerge in other cities or rural areas. Third, the study sample was not representative of all low-income racial and ethnic minority residents of public housing, further constraining the broader applicability of the results. Additionally, key contextual factors, such as neighborhood characteristics, access to public transportation, and broader systemic inequities, were not included in the analysis but could play a critical role in shaping telehealth utilization.

### 4.4. Future Research

Future research should address these limitations by employing longitudinal designs to better understand the dynamic and evolving nature of telehealth adoption over time. Tracking changes in telehealth attitudes and utilization in response to policy shifts, technological advancements, or targeted interventions could provide deeper insights into causal relationships. Moreover, qualitative studies are essential to explore the nuanced experiences and barriers faced by this intersectional population, adding depth to the quantitative findings and uncovering factors that may not be captured by surveys or structured data collection. Larger and more representative sample sizes should also be prioritized to enhance the generalizability of findings. Expanding the scope of research to include additional variables such as neighborhood characteristics, access to public transportation, and mental health indicators (e.g., depression) could provide a more comprehensive understanding of the factors influencing telehealth use. Furthermore, incorporating measures of discrimination and mistrust—common barriers in healthcare access for marginalized populations—would help to elucidate their potential impact on telehealth attitudes and behavior. By addressing these gaps, future studies can build on the present findings to inform more effective policies and interventions aimed at promoting equitable telehealth adoption among underserved communities.

## 5. Conclusions

In conclusion, this study underscores the importance of addressing structural barriers to increase telehealth use among low-income racial and ethnic minority residents of public housing. While telehealth attitudes, such as expectancy and self-efficacy, were not significant predictors of utilization, demographic and systemic factors played a more prominent role. These findings suggest that structural interventions, rather than solely focusing on individual attitudes, are necessary to bridge telehealth disparities and promote equitable access to healthcare in underserved communities.

## Figures and Tables

**Figure 1 healthcare-13-00286-f001:**
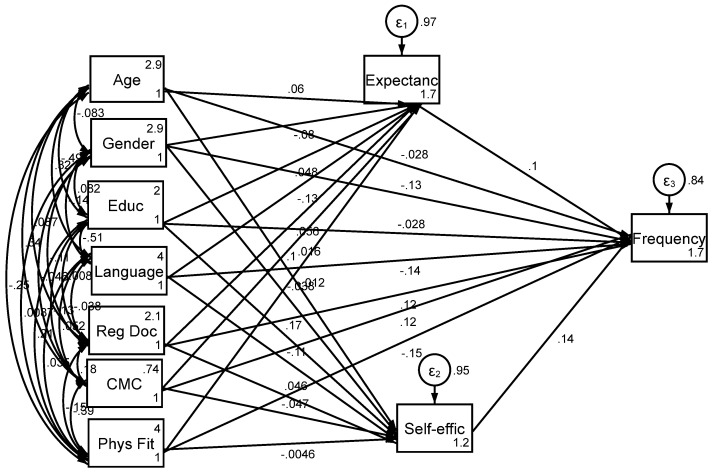
Summary of the structural equation model with telehealth frequency as the outcome. Note: CMC: chronic medical condition; Reg Doc: having a regular provider; Educ: own educational attainment; Language (Spanish); Gender: male gender; Physical Fit (SF12 Score); Expectanc: telehealth expectancy (expected benefits); Frequency: telehealth utilization frequency; Self-effic: self-efficacy to use telehealth.

**Table 1 healthcare-13-00286-t001:** Descriptive statistics for demographic variables (n = 252).

	**N (%)**
Gender	
Male	64 (25)
Female	188 (75)
Education	
8th grade or less	99 (39)
Some high school, no diploma	56 (22)
Graduated from high school or have a GED equivalent	66 (26)
Associate degree or some college/Bachelor’s Degree	31 (12)
Language	
English	65 (26)
Spanish	187 (74)
Primary Care Provider	
No	45 (18)
Yes	207 (82)
Chronic Conditions	
0	122 (48)
1	47 (19)
2	39 (15)
3 or more	44 (18)
How many times have you used telehealth in the past 12 months?	
0	133 (53)
1	50 (20)
2	37 (15)
3	19 (7)
4 or more	13 (5)
	**Mean (SD)**
Age (Yrs)	45 (15)
Physical Mobility	9 (2)
Attitude Toward Telehealth Care Utilization (Self-Efficacy)	2.151 (0.11)
Attitude Toward Telehealth Care Utilization (Expectancy)	2.381 (0.12)
Number of Chronic Conditions (scale of 0–8)	1 (2)

**Table 2 healthcare-13-00286-t002:** Bivariate correlations between telehealth utilization and other study variables.

	1	2	3	4	5	6	7	8	9	10
1. Age	1.00									
2. Gender	−0.08	1.00								
3. Education	−0.49 **	0.08	1.00							
4. Language	0.32 **	−0.14 *	−0.51 **	1.00						
5. Physical Mobility Limitation	−0.25 **	0.01	0.21 **	0.03	1.00					
6. Number of Chronic conditions	0.34 **	−0.05	−0.13 *	0.05	−0.39 **	1.00				
7. Primary Care Provider	0.09	−0.11	−0.01	−0.04	−0.15 **	0.18 **	1.00			
8. Attitude Telehealth Care Utilization (Expectancy)	0.01	−0.07	0.07	−0.13 *	−0.04	0.04	0.08	1.00		
9. Attitude Telehealth Care Utilization (Self-Efficacy)	−0.02	−0.02	0.18 *	−0.16 **	0.01	−0.03	0.05	0.76 **	1.00	
10. Telehealth Care Utilization (Frequency)	0.04	−0.14 *	0.03	−0.15 *	−0.22 **	0.20 **	0.20 **	0.25 **	0.24 **	1.00

Note: ** Correlation is significant at the 0.01 level (two-tailed), and * correlation is significant at the 0.05 level (two-tailed).

**Table 3 healthcare-13-00286-t003:** Summary of the structural equation model with telehealth frequency as the outcome.

	B	SE	95%	CI	*p*
**Telehealth Expectancy**					
Age (Yr)	0.060	0.076	−0.089	0.208	0.432
Gender (Men)	−0.080	0.063	−0.203	0.043	0.201
Education	0.048	0.080	−0.109	0.205	0.548
Language (Spanish)	−0.132	0.074	−0.276	0.013	0.074
Regular Doctor	0.056	0.064	−0.069	0.180	0.381
Chronic Disease	0.016	0.071	−0.123	0.154	0.826
Physical Fitness	−0.012	0.070	−0.149	0.125	0.867
Intercept	1.678	0.584	0.533	2.823	0.004
**Telehealth Self-Efficacy**					
Age (Yr)	0.104	0.075	−0.043	0.251	0.164
Gender (Men)	−0.038	0.062	−0.160	0.084	0.545
Education	0.172	0.079	0.018	0.326	0.029
Language (Spanish)	−0.112	0.073	−0.254	0.031	0.126
Regular Doctor	0.046	0.063	−0.077	0.170	0.463
Chronic Disease	−0.047	0.070	−0.184	0.090	0.504
Physical Fitness	−0.005	0.069	−0.140	0.131	0.947
Intercept	1.153	0.581	0.013	2.292	0.047
**Telehealth Utilization (Frequency)**					
Telehealth Expectancy	0.102	0.090	−0.075	0.280	0.258
Telehealth Self-Efficacy	0.142	0.091	−0.036	0.320	0.118
Age	−0.028	0.071	−0.168	0.111	0.691
Gender (Men)	−0.128	0.059	−0.243	−0.013	0.029
Education	−0.028	0.076	−0.177	0.121	0.712
Language (Spanish)	−0.135	0.069	−0.270	0.000	0.050
Regular Doctor	0.121	0.059	0.005	0.237	0.041
Chronic Disease	0.125	0.066	−0.004	0.254	0.058
Physical Fitness	−0.149	0.065	−0.276	−0.022	0.021
Intercept	1.748	0.547	0.676	2.820	0.001

## Data Availability

The original contributions presented in this study are included in this article. Further inquiries can be directed to the corresponding authors.

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
