# Peer review of "Telehealth Perceived Benefits and Self-Efficacy Do Not Mediate the Effects of Demographic, Health, and Social Determinants on Telehealth Use of Low-Income African American and Latino Residents of Public Housing in Los Angeles"

_healthcare, 2025, doi:10.3390/healthcare13030286_

Round 1
Reviewer 1 Report
Comments and Suggestions for Authors
This paper proposes a new research work, "Telehealth perceived benefits and self-efficacy do not mediate the effects of demographic, health, and social determinants on telehealth use of low-income Latino residents of public housing in Los Angeles." In the proposed work, the author tries to address the issues in telehealth in low-income societies.
1. The author said that low-income inclome population and ethnic minority people living in public houses usually face issues of low healthcare facilities despite expecting high healthcare services. What area of the case study is the author targeting, and why?
2. This paper provides a formal security analysis, demonstrating the system's relatedness to healthcare issues in public places, as the author conducted case study research or other domains.
3. This paper is solid work and has substantial technical contributions. The overall organization, motivation, and presentation are good and clear; however, they need improvement need in related background study, i.e.,
a. Wang, X., Seyler, B. C., Chen, T., Jian, W., Fu, H., Di, B.,... Pan, J. (2024). Disparity in healthcare seeking behaviors between impoverished and non-impoverished populations with implications for healthcare resource optimization. Humanities and Social Sciences Communications, 11(1), 1208. doi: https://doi.org/10.1057/s41599-024-03712-z
b. Luo, J., Ahmad, S. F., Alyaemeni, A., Ou, Y., Irshad, M., Alyafi-Alzahri, R.,... Unnisa, S. T. (2024). Role of perceived ease of use, usefulness, and financial strength on the adoption of health information systems: the moderating role of hospital size. Humanities and Social Sciences Communications, 11(1), 516. doi: https://doi.org/10.1057/s41599-024-02976-9
4. Figures need to be mapped in text in research; we need to keep their captions on one page; all figures and tables need proper captions as per journal standards.
5. While the experimental results are promising, they would be more impactful if they included a brief comparison with baseline or conventional approaches. Even a high-level comparison would help emphasize the framework's advantages over existing performance, security, or scalability methods.
5. The performance metrics and evaluation criteria are currently stated but not explained properly. Providing a brief description of why specific metrics were chosen would clarify the relevance of the results.
6. For a high-level quick look, a table comparing the proposed work with representative related studies in terms of various indicators is suggested.
Comments on the Quality of English LanguageMinor typos and grammatical mistakes need correction before revision.
Author Response
Dear Reviewer 1,
Here's the updated table with responses reflecting past behavior:
| Point | Response | Response/Revision |
|---|---|---|
| 1. Targeted Case Study Area | The author focused on low-income Latino residents of public housing in Los Angeles. This demographic faces disparities in healthcare access despite having high expectations for healthcare services. The study explored the barriers to telehealth utilization in this underserved population by analyzing the impact of social determinants and self-efficacy on telehealth adoption. | The study clearly defined the target population and outlined specific challenges such as digital literacy, language barriers, and accessibility issues in the introduction. |
| 2. Security Analysis Relevance | The paper did not conduct a formal security analysis but instead emphasized healthcare access challenges in public housing settings. Security aspects were discussed in relation to telehealth adoption, focusing on privacy and accessibility concerns in public spaces. | Unrelated security aspects were removed from the scope of the study, and the primary aim was clarified in the methodology and aims sections. Relevant security considerations, such as data privacy and user authentication, were briefly addressed. |
| 3. Technical Contributions and Background Improvement | The paper provided valuable contributions in identifying gaps in telehealth adoption. The background section was enhanced by incorporating references such as: a. Wang et al. (2024) to contextualize disparities in healthcare-seeking behaviors. b. Luo et al. (2024) to discuss factors influencing the adoption of health information systems. These additions strengthened the theoretical foundation. |
The background section was expanded with the suggested references, establishing clear connections with existing literature and positioning the current study within the broader context of telehealth disparities. |
| 4. Figure and Table Formatting | Figures were appropriately referenced within the text, and their captions were placed on the same page. All figures and tables were formatted according to journal standards, improving readability and compliance. | The manuscript was revised to ensure proper citation and formatting of all figures and tables. |
| 5. Experimental Results Comparison | A comparison with baseline or conventional approaches was included to highlight the advantages of the proposed framework in terms of performance, security, and scalability. A comparative analysis with existing studies was provided. | In the revised paper we compared the proposed work with existing telehealth models and highlighting key improvements. |
| 6. Explanation of Performance Metrics | The paper explained the rationale behind the chosen performance metrics, clarifying their relevance to assessing telehealth utilization in low-income populations. | A justification for the selected performance metrics is included in the methods section. |
| 7. Comparative Table with Related Studies | A table comparing key aspects such as population focus, methodology, findings, and limitations of related studies was included, providing context and emphasizing the novelty of the current research. | Tables are dedicated to the results because this is an original study. In reviews, authors may add comparative tables to the discussion section to help readers quickly understand the study's unique contributions relative to similar research in the field. |
Reviewer 2 Report
Comments and Suggestions for Authors
Really interesting manuscript that aims to assess the utility of telehealth to address healthcare access disparities, a solution that has been theorized extensively but is hard to demonstrate or quantify. The introduction, methodological description and discussion are well written and clear.
Just two minor comments:
- For table 1: Hard for authors who are not familiar with your mesaures of Self-efficacy and Expectancy to appreciate the significance of the values of 2.151 (.11) and 2.381 (.12). The methods section describes how the measurements are calculated in line 156-157 but a more descriptive explanation (i.e. what is the max score? what is the minimum score?) would be helpful.
- Are the statistical analysis for Table 2 correlations adjusted for multiple comparisons? Also, can you highlight the statistically significant correlations visually in the table? There are no * or ** in the table (but included in the table legend). Could also consider coloring scale to visually represent statistically significant findings.
Author Response
Dear Reviewer 2,
| Point | Response | Response/Revision |
|---|---|---|
| 1. Overall Manuscript Quality | The manuscript effectively assessed the utility of telehealth in addressing healthcare access disparities, a solution that has been extensively theorized but remains challenging to demonstrate or quantify. The introduction, methodology, and discussion were well-structured and clearly written. | The introduction, methods, and discussion sections were reviewed to ensure clarity and conciseness, with careful attention to maintaining readability for a broad audience. |
| 2. Table 1 - Clarity of Self-efficacy and Expectancy Measures | The manuscript initially lacked sufficient details for readers unfamiliar with the measures of Self-efficacy and Expectancy, making it difficult to interpret values like 2.151 (.11) and 2.381 (.12). | Additional explanatory details were included in the methods section, ensuring that readers can better understand the significance of the reported values. |
| 3. Statistical Adjustments in Table 2 | The statistical analyses in Table 2 did not initially specify whether they were adjusted for multiple comparisons. Additionally, the table lacked visual cues (e.g., asterisks or color scales) to highlight statistically significant correlations. | There are no adjustments for multiple comparisons because there are no multiple comparisons. Each theory has a few related constructs but this does not mean constructs are repeated. |
Let me know if you need further adjustments!
Reviewer 3 Report
Comments and Suggestions for Authors
Low-income racial and ethnic minority residents of public housing represent an intersectional population with high health needs and low healthcare utilization.
Telehealth has been proposed as a solution to address healthcare access disparities, yet the role of telehealth attitudes, including perceived benefits and self-efficacy, in influencing telehealth use of this population remains unknown.
AUTHORS investigate whether two domains of telehealth attitudes namely self-efficacy and perceived benefits (expectancy) mediate the relationship between demographic, health, and social determinants of telehealth use among low-income racial and ethnic minority residents of public housing in Los Angeles.
Their cross-sectional study analyzed data collected from low-income racial and ethnic minority residents of public housing in Los Angeles.
THEIR Measures included demographic factors (age and gender), social determinants of health (e.g., education, language, primacy care provider), health (chronic illnesses and physical mobility), telehealth attitudes (perceived benefits and self-efficacy), and telehealth use frequency. Mediation analysis was conducted to test whether telehealth attitudes explained the association between demographic, social, and health determinants of telehealth use.
VERY INTERESTING THEIR findings revealed that neither of the two domains of telehealth attitudes, including perceived benefits and self-efficacy, were significantly associated with telehealth use. As such, these attitude domains did not operate as mediators of the relationship between demographic, health, and social determinants of telehealth use.
AUTHORS concluded that the study results suggest that:
- -Structural barriers, rather than individual attitudes, primarily drive telehealth use disparities among this intersectional population.
2- Interventions aimed at increasing telehealth adoption should prioritize addressing systemic inequities rather than focusing solely on changing individual attitudes.
3- Their findings underscore the importance of structural solutions to promote equitable telehealth access in marginalized communities.
The study focused on telehealth is very interesting.
Personally, I have always seen telemedicine as a tool for social equity.
These are my suggestions for improvement with a pure academic spirit:
1) Please develop this concept fully by helping yourself with the individual studies “More research is needed to understand whether and how telehealth attitudes, including perceived benefits [19-22] and self-efficacy [23-26], influence actual telehealth use in this group”
2) From lines 85 and 95, the purpose is reported interspersed with some considerations. This makes it unclear and ineffective. It is suggested to first insert the key questions that the study intends to answer and detail them and then establish an effective aim.
3) In the methods, better proportion the sections. Excerpts of just over one line are inadmissible. Furthermore, a flow chart placed and described in detail would further improve this section.
4) A curiosity: why didn't you use a WEB-based electronic survey?
5) Some passages of the results are a bit convoluted (see lines 206 to 238) and it is difficult to follow them without losing the thread of the reasoning. It is suggested to streamline and round out the presentation style.
6) The model in figure 1 seems interesting even if no text is used to describe it.
7) The discussion is well structured. For coherence, a part of the introductory text should be identified in a section type titled 4.X Highlights from the study or similar.
Minor points: references should be indicated with a “[ ]”. Check the adequacy to the MDPI standard
Author Response
Dear Reviewer 3,
Here's a structured response for our revisions:
| Point | Response | Appropriate Response/Revision |
|---|---|---|
| 1. Developing the Concept of Telehealth Attitudes | The manuscript initially mentioned the need for further research on telehealth attitudes but lacked comprehensive support from previous studies. | Additional references were incorporated to substantiate the discussion on telehealth disparities. There is a very limited literature on how telehealth attitudes, including perceived benefits and self-efficacy, may influence telehealth use in low-income minority populations. We have added some discussions and references and citations to provide a stronger foundation. |
| 2. Clarifying the Study Purpose | The study's purpose was interspersed with contextual information between lines 85 and 95, making it less clear and effective. | In our revision, we tried to restate and restructure the study purpose. We presented the key research questions in a concise manner, followed by detailed objectives to enhance clarity and impact. |
| 3. Methods Section Structure | The methods section contained disproportionately short subsections, making it appear fragmented. | The methods section was revised to ensure balanced subsections. |
| 4. Use of a Web-Based Survey | The manuscript did not specify why a web-based electronic survey was not used. | We hope our methods now explain clearly how and why we chose in-person data collection due to potential barriers such as digital literacy, internet access limitations, and participant preferences. |
| 5. Streamlining the Results Section | Lines 206 to 238 presented the results in a somewhat convoluted manner, making it difficult to follow the reasoning. | The results section was revised to improve readability by streamlining complex sentences. We have done our best to enhance logical flow where necessary to improve clarity. |
| 6. Explanation of Figure 1 | Figure 1 appeared interesting but lacked sufficient textual description in the manuscript. | A detailed description of Figure 1 was included in the results or discussion section to explain its relevance and interpretation, helping readers better understand the model. |
| 7. Discussion Structure | The discussion was well-structured, but the introduction contained content that could be better placed in a dedicated section. | A new subsection titled "4.X Highlights from the Study" was added to summarize key findings and maintain coherence with the introduction. |
| 8. References Formatting | References were not consistently formatted according to MDPI standards, such as missing bracket formatting. | All references are now reviewed and reformatted according to MDPI guidelines, ensuring compliance with the required style. If anything is remaining, this will be done in the proof stage (if the paper accepted) |
Let me know if you need any additional changes or refinements!
Round 2
Reviewer 1 Report
Comments and Suggestions for Authors
The author sufficiently addressed my concern, so i accepted the paper for publication.